# The Promotion of Mental Health and Prevention of First-Episode Psychosis: A Pilot and Feasibility Non-Randomised Clinical Trial

**DOI:** 10.3390/ijerph20227087

**Published:** 2023-11-20

**Authors:** Lucia Santonja Ayuso, Antonio Ruiz-Hontangas, José Javier González Cervantes, Concepción Martínez Martínez, Eva Gil Pons, Sonia Ciscar Pons, Laura Andreu Pejó, José Vicente Carmona-Simarro

**Affiliations:** 1Department of Nursing, Faculty of Health Sciences, Universidad Jaume I, 12006 Castellón, Spain; al400276@uji.es (L.S.A.);; 2Department of Nursing, Faculty of Health Sciences, Universidad Europea de Valencia, 46010 Valencia, Spain

**Keywords:** social stigma, psychosis, mental health, mental disorders, school teachers, adolescent

## Abstract

Background: Mental-health-related stigma prevents active help seeking and therefore early therapeutic approaches and the recovery of functionality. National and international agencies recommend the implementation of prevention and mental health promotion programs that support the elimination of stigma in the classroom, since most mental health problems usually start in the adolescent stage. In view of the evidence that teachers present stigmatizing attitudes towards mental health, it has been considered as convenient to carry out an anti-stigma program with the main objective of evaluating the impact of an intervention based on the education and promotion of mental health, aimed at teachers and counsellors of a secondary school. The specific objectives were to get to know which were the most stigmatising attitudes that prevailed in the sample before and after the intervention; to evaluate the knowledge of the teaching staff and counsellors on psychosis before the intervention; to analyse correlations between clinically relevant variables; and assess whether this programme was beneficial and feasible for alphabetising counsellors/teachers of educational centres on stigma and FEP. Methods: This was a non-randomised clinical trial in which a nursing intervention was performed. Tools: a psychosis test (pre), Stigma Attribution Questionnaire (AQ-27) (pre-post), and satisfaction survey (post) were used. The inferential analysis included the Wilcoxon and the Pearson Correlation Test. Results: In the sample (*n* = 22), the predominant stigmatising attitude was “Help”. The *p*-values obtained in the Wilcoxon Test were statistically significant, except for “Responsibility” and “Pity”. The following constructs of interest were faced: “Fear”–“Age” and “Professional experience”; and “Help”–“Psychosis test”. Conclusions: Despite the scores obtained in “Responsibility” and “Pity”, the intervention was useful for reducing stigma in the sample. Implications for the profession: There are adolescents who have suffered stigma from their teachers, and consequently have minimized their symptoms and not asked for help. For this reason, we implemented a nursing intervention based on the education and promotion of mental health, with the aim of expanding knowledge and reducing stigma. In fact, this intervention, which we carried out on high school teachers, managed to reduce the majority of stigmatizing attitudes measured on the stigma attribution scale.

## 1. Introduction

Adolescence is a concept that is difficult to categorise and comprises a transition period from infancy to adulthood, in which there are important changes at a physical, cognitive, and emotional level, making it a phase filled with energy and opportunities, but also with risks and vulnerabilities [1]. In fact, the literature suggests accompanying the adolescent in the academic, social, and healthcare environments, for them to create their own identity in a positive way through the promotion of mental health and prevention of the pathology [2,3].

For that, and following the latest international recommendations of the World Health Organisation (WHO) published in its Comprehensive Mental Health Action Plan 2013–2020 [4], the “Mental Health Strategy of the National Health System 2022–2026” [5] was recently published in Spain, in which the importance of prioritising mental health in all environments was emphasised, and it focuses on the promotion and prevention of mental disorders and action against stigma.

Stigma is an extremely devaluing concept that, from the socio-cognitive perspective, is composed of stereotypes, prejudices, and discrimination that are transmitted intergenerationally and are perpetuated through the media [6,7]. Consequently, this not only allows society to fear and avoid people with a mental disorder, but also for them to minimise the clinic, treatment, and their recovery due to shame [8,9,10,11].

In fact, there is evidence that states that adolescents with mental disorders have experienced stigmatisation from their peers, families, and teachers, allowing negative emotional reactions, self-discriminatory attitudes, and isolation [12,13]. This fact is particularly striking, since teachers can be a unique figure for providing the necessary support in these cases, as they spend most of their time with their students, and to do so, they must eradicate the levels of stigma (if they have them) and increase their knowledge on mental health [14].

At the same time, and despite the multiple instructions for carrying out specific programmes to increase knowledge on mental health and reduce stigma associated with the health, social, and educational systems, there is scarce scientific evidence on the latter, since the majority of studies made have focused on students of different categories or on healthcare professionals [15,16,17,18]. Moreover, the few programmes performed on teaching staff have dealt almost exclusively with disorders like anxiety and depression [6,19], leaving aside important disorders like psychosis, due to its high rates of Disability-Adjusted Life Years (DALY) [3,5].

Given that psychosis, or First-Episode Psychosis (FEP) if it occurs for the first time, usually debuts from age 15 and its onset is progressive and insidious, where there is an alteration of reality [3,20,21], it would be interesting to perform a psychoeducational programme based on the Vulnerability-Stress model (as it is the most accepted model today for the explanation of the risk factors associated with the onset of psychosis) to increase knowledge on mental health and FEP on teachers and counsellors of a secondary school in order to reduce the associated stigma [3,22]. Some of the ways of carrying this out that have been documented in the last few years are psychoeducation and the live interpersonal exchange of people with experiences on mental health disorders [23], although similar effects have been found in the latter when people contact virtually (videos, web interactions) [24,25].

For this reason, our intervention study has the general objective to evaluate the impact of an intervention based on the education and promotion of mental health and anti-stigma, aimed at teachers and counsellors of a secondary school located in Spain. The specific objectives were to get to know which were the most stigmatising attitudes that prevailed in the sample before and after the intervention; to evaluate the knowledge of the teaching staff and counsellors on psychosis before the intervention; to analyse correlations between clinically relevant variables; and assess whether this programme was beneficial and feasible for alphabetising counsellors/teachers of educational centres on stigma and FEP.

## 2. Material and Methods

### 2.1. Design

This was a quasi-experimental (pre and post-test) clustered pilot study carried out in a secondary school located in Spain. For this reason, the TREND checklist (Transparent Reporting of Evaluations with Nonrandomized Designs) was used [26].

### 2.2. Participants and Recruitment

The sample was composed of teachers and counsellors of a secondary school located in Spain. It was selected out of a simple random sampling between the secondary schools to which we had access that belong to the department in Spain. The software programme Excel^®^ (Version.18.0, Excell, Office 365, Microsoft Corporation, EEUU) was used to perform this task.

This type of population was chosen since they are considered as communitary workers that spend most of their time with their students (being able to closely observe any behavioural changes over time) aged from 15 to 18 (starting period of FEP, for the most part).

For recruitment, in the first place, a random sampling of the schools that belonged to the health area was carried out. In second place, counsellors of the selected secondary school were contacted in order to participate and were sent an email with the objectives and programme of the activity, together with an informed consent form that had to be signed by the director of the centre. After establishing the pertinent inclusion and exclusion criteria, another informed consent form was given to the participants and was signed by all of them.

All the participants that complied with the selection criteria were included, then performing intentional-type sampling.

### 2.3. Inclusion and Exclusion Criteria

Inclusion criteria: To work as a teacher and/or counsellor at the moment of the programme presentation, to have Spanish as mother tongue, to develop the teaching activity full-time, to voluntarily participate, and to be public, semi-private, and private secondary schools.

Exclusion criteria: Not having signed the informed consent, to be part-time teachers and/or teachers on sick leave, and to be nursery schools and public and/or private universities.

### 2.4. Assessment Instruments

#### 2.4.1. Stigma

The tool used for the assessment of stigma was the Attribution Questionnaire (AQ-27) (created by Weiner and collaborators) “(1988)” in its Spanish version, which is composed of 27 items (AQ-27-E). The evaluation was carried out by Muñoz and collaborators “(2015)”, obtaining a reliability of 0.855 (Cronbach’s alpha) for the scale total and presenting appropriate psychometric features for the Spanish population [27].

It consists of a self-administered questionnaire comprised of 27 items that are evaluated using a Likert-type scale (1 corresponding to “not at all” and 9 to “extremely”) whose administration time is 10–20 min. The results of 9 subscales are obtained: “Responsibility” (the person is responsible for controlling their disorder), “Pity” (to feel empathy), “Anger” (to feel wrath), “Danger” (to believe they are a threat), “Fear”, “Help” (willingness to assist a person with psychiatric-type problems), “Coercion” (obligation to receive treatment), “Segregation” (exclusion), and “Avoidance”. It was performed at the beginning and after completing the intervention.

It consists of 10 questions about prevalence, aetiology, course, and the early detection of psychosis. It is based on the scientific literature and was elaborated by the author and subsequently read, checked, and structured by a total of 10 experts from the FEP Functional Unit using the Delphi technique with the aim of reaching a consensus with the experts. The range of results was from 0 to 10 points; it was performed at the beginning of the intervention.

#### 2.4.2. Satisfaction Survey

The satisfaction survey adapted from Feixas and Vilaplana [28] was used: an overall rating of the intervention (teachers, material, methodology, and contents) by the means of a numeric scale (from 1 to 10). The internal consistency for the Spanish version is between 0.70 and 0.90, which is satisfactory [29]. It was performed at the end of the intervention.

### 2.5. Intervention

A training workshop on psychosis and general mental health was carried out in order to raise the awareness of the teaching staff and counselling department. It was a psychoeducational-type intervention that was fundamentally inspired by the content of other guides, recommendations, and renowned entities in the field of mental health in Spain [5,12,30,31]. The intervention was carried out by a mental health specialist nurse.

The modules were designed to confront stigma regarding mental disorders and to identify possible risk factors and behavioural changes, so that a subsequent preventive effort could be made. Within the health Education and Promotion framework, audiovisual materials, games, websites, and a recorded interview with a person that suffered from schizoaffective disorder were used, since they are resources that have shown to be effective in reducing stigma [32,33]; in addition, it seems like the latter is the one that has made it possible to obtain better results [34,35]. The activity was performed in the educational centre and was structured in two big blocks, “stigma” and “psychoeducation in psychosis”. The contents covered included: stigma (definition, causes, and consequences; reflection and eradication), risk factors related to mental disorders, conflict resolution, substance abuse prevention, the early detection of alarm symptoms, referral to specialised services, treatment (pharmacological and non-pharmacological), and recovery in school, altogether with group dynamics. Its approximate duration was 3 h in total, divided into three sessions (one each week).

The intervention was structured from August to December 2021 and was implemented at the end of February 2022.

### 2.6. Data Analysis

All the statistical analyses were performed with IBM SPSS Statistics version 23. The level of significance was established with a *p* value of ≤0.05 and a confidence index of 95%.

Univariate analysis. The baseline sociodemographic and clinical features of the group were described using measures of central tendency for quantitative variables, and total frequency table and percentage for categorical variables.

Bivariate analysis. The parametric statistic Pearson’s Correlation was used to analyse the correlation between two quantitative variables: The degree of relation was established with the following criterion: *r* higher than 0.5 (strong correlation), *r* between 0.5 and 0.25 (moderate correlation), and *r* lower than 0.5 (weak correlation).

The Wilcoxon test was used to assess the effectiveness of the intervention.

## 3. Results

### 3.1. Characteristics of Patients

A total of 22 participants (18% men and 82% women) were recruited for this study and completed the pre- and post-intervention questionnaires. The sample profile was mostly women aged 47.96 ± 10.12 years, with undergraduate studies and a professional experience of 18.64 ± 10.34 years. The rest of the demographic characteristics are detailed in Table 1.

### 3.2. Means of the Pre- and Post-Intervention AQ-27 Scores

Table 2 shows the means obtained for each stigmatising attitude, identifying “Coercion” as the predominant stigmatising variable before and after the intervention. It also shows the results of the *p*-values obtained in the Wilcoxon Test among the mean scores of each stigmatising attitude that comprise the AQ-27 scale before and after the intervention.

It is observed that there are statistically significant differences in all the stigmatising variables (in other words, these differences are not attributed to fortuity) except for “Responsibility”.

### 3.3. Correlations “Age”–“AQ-27 Pre-Intervention” and “Experience”–“AQ-27 Pre-Intervention”

These correlations can be seen in Table 3, with a statistically significant and inversely proportional relation, moderate in nature, between “Age” and “Fear”; that is to say, the older the sample, the less fearful they are of people with mental disorders. This result coincides with the correlation between “Professional experience” and “AQ-27 pre-intervention” (Table 4).

### 3.4. Correlations of “Psychosis Test”–“Age” and “Psychosis Test”–“Professional Experience”

As seen in Table 5, as age and professional experience increase, knowledge on psychosis decreases. This result had no statistical significance.

### 3.5. Psychosis Test

The mean score obtained before the intervention was 4.89 out of 10. Table 6 shows correlations between “Psychosis test” and “AQ-27 pre-intervention”. The statistically significant and directly proportional relation between knowledge on psychosis and the item “Help” is highlighted.

### 3.6. Course Rating

Points are shown in Table 7. There were no side effects.

### 3.7. Semi-Structured Interview

Since the end of the program, counsellors reported 11 cases referred to the health system, 9 of which required assessment and follow-up by the Child and Adolescent Mental Health Unit: 5 boys and 4 girls between 13 and 16 years due to behavioural alterations and anxious symptoms.

## 4. Discussion

From what is known, this is the first study that has explored the effect of a psychoeducational programme on mental health and FEP aimed at teachers and counsellors in a secondary school in Spain. The outcomes showed that the intervention was useful for increasing knowledge on mental health and reducing the stigma associated with this type of population, given the statistically significant differences found in the average scores of the AQ-27 variables pre- and post-intervention.

This result could be explained by the approach of the programme in the promotion and prevention of mental disorders, the implementation of group dynamics, and the audiovisual sources used, since these are tools that have been scientifically endorsed by other stigma reduction programmes [36,37]. However, in line with a similar study from 2021, it should be noted that the intervention had to be adapted prior to the schedule of the sample, making it impossible to randomly sample participants and perform a posterior follow up [38].

It should be highlighted that, in the present study, the sample started from low levels of stigma, as opposed [7,25,39], in which high scores were found in all studied stigma dimensions. This difference may be due to the type of population that teachers treat (minors) in contrast with the adult population when it comes to previous studies (students and healthcare professionals).

However, the scores obtained in the “Pity”, “Help”, and “Coercion” subscales were in accordance with other studies [16,26]. On the one hand, and taking into account the results of “Pity” and “Help”, it could be suggested that teachers and counsellors empathise with the person suffering from a mental disorder and would be willing to help them; on the other hand, the results could indicate that our sample considers that these people are dependent and incapable of making decisions, and therefore would need help, in line with other research like that from Hamilton and Eissa [40,41]. In fact, the latter would coincide with other studies that suggested that nursing students prioritised the fulfilment of the medical treatment because they considered the person with a mental disorder to be incapable of making decisions regarding their own health and avoiding their participation in decision making suggested [15,16,42].

As for “Coercion”, it should be noted that there were no high scores in other related subscales like “Danger”, “Anger”, “Fear”, “Avoidance”, or “Segregation”, in contrast to what has been obtained in other studies [16,42,43]. In our sample, this fact could be explained by different reasons: firstly, due to the scarce knowledge that our sample presented on psychosis in general, as the scores obtained in the Psychosis Test showed; and, secondly, due to the context of action in which our participants are, since they are responsible for the adolescents within the centre, and must comply with the parents’ indications regarding medical issues (for example, knowing allergies and/or administering medication).

Although it is true that “Coercion” continued to be the most stigmatising attitude after completing the programme, its statistically significant differences, along with the other items, stated that the contents of the programme were useful for working on this stigmatising attitude in this type of population, then answering our research hypothesis.

That said, the only item against which the programme was not effective was “Responsibility”; that is to say, participants continued to believe that the person was responsible for the onset and maintenance of their illness. The study of Hunter [44], which was in the same line, suggested, altogether with Hinshaw [45], to focus on the cause of the mental disorder in the recapitulation theory in order to increase empathy and decrease guilt [16]; in other words, by placing the responsibility for the onset of psychosis on exclusively genetic factors, society’s behaviour toward a person with psychosis would be more compassionate and supportive. However, the paradigm used in this programme is “Vulnerability-stress” by Zubin and Spring [22], since this multicausal model is the most accepted nowadays [3], and, in this case, one of the risk factors that stood out from the rest within the group was substance use.

That is why we believe that an opportunity to improve the scores obtained in this subscale would be adding specific material or performing a complementary programme that treated the prevention and consequences of the consumption of toxic substances, since we consider that this also presents stigma and disinformation because it can cause substance-induced psychosis, but other factors like addiction and the severity of consumption should be taken into account, as Fiorentini [46] stated in their study.

Regarding the correlations performed, we find interesting to mention the statistically significant relation between age and work experience compared with “Fear”, since it was the only statistically significant negative variable, in contrast with that obtained in the study by Tooby and Cosmides [47], in which “Fear” increased proportionally as age increased. Taking into account the scarce knowledge of the sample regarding psychosis and mental disorder, and that all scores of the AQ-27 subscales could be further reduced, we emphasise the importance of actively involving the healthcare and educational sectors in the needs and requests of adolescents regarding mental health [16,48,49,50,51].

Even so, ahead of the severe consequences of stigma and the promising result of the programme (considering that counsellors have confirmed the practical application of this program by making an appropriate derivation), its performance is suggested as a mandatory and educational part of the curricular schedule of the centre. It would get both teachers and counsellors to increase their theoretical–practical knowledge on mental health, to acquire skills on handling possible difficult situations, and even to develop the necessary skills to apply actions against stigma and early detection from the classroom.

### 4.1. Strengths and Limitations

Since the study was carried out voluntarily in real life in La Ribera health department, it was not possible to randomise the participants, there could not be a control group, the sample could not be randomised given the prior need of planning lessons, and the intervention content had to be adapted to the available time of the sample, like in other studies [14,19]. However, its duration was similar to the one of another alphabetisation study about mental health that also had promising outcomes (although related to depression) [51], regardless of not being able to perform a posterior follow up from the centre, as has happened in similar research [14]. Therefore, in the interpretation of the results, the voluntariness bias must be considered, and it may be that the scores in the AQ-27 are higher than those obtained, since the teachers most interested in mental health were those who attended the program.

In any case, we are satisfied with the results obtained with this programme due to the reduction in the scores of stigmatising attitudes. Together with the contribution of knowledge on mental health, the possibility that an alteration can be identified early and more positively approached by this population, this programme will directly impact health and recovery of adolescents.

However, in order to make sure it happens, studies that include a larger number of participants and include a follow-up on stigma levels, psychosis knowledge, and number of cases identified would be required. Other detected limitations were time and lack of research in compliance with our line of results, making it impossible to compare results over time and with other research.

### 4.2. Future Lines

Given the promising data of this Mental Health Nursing psychoeducational-type intervention, its performance in other educational centres is suggested, taking into account that the sample should be more homogeneous and representative, preferably scheduled within work hours and with the possibility of obtaining further official certification. Perhaps this way, the sample size can be increased. Or, indeed, adding this type of training to the curricular records of teachers and counsellors of the centre and give the option to do it both onsite and online.

The importance of performing a posterior follow-up of the clinical variables of interest must be emphasised to be able to answer if having knowledge on psychosis reduces the attribution stigma, as well as using other complementary scales to enrich the result obtained.

## 5. Conclusions

The impact of a nursing intervention on mental health stigma in teachers and counsellors of a secondary school was positive, given the statistically significant differences observed in the change of average scores on stigmatising attitudes of the AQ-27 scale pre- and post-intervention. The stigmatising attitude that prevailed in the sample before and after the intervention was “Coercion”. It was objectified that knowledge on psychosis before the intervention was scarce, which possibly influenced wrong psychosis beliefs. The correlations made stated that, the older the person or the more experience in the sector, the lower fear towards the mental disorder is, and complementary qualitative studies should be carried out to learn more about why this does not happen with the rest of variables. Given the overall result of all the scores obtained, including the satisfaction scale, we consider that the programme was helpful and feasible for alphabetising counsellors and teachers of educational centres on stigma and First-Episode Psychosis.

## Figures and Tables

**Table 1 ijerph-20-07087-t001:** Sociodemographic data of the sample.

Variable	%	*AF*
Sex		
Male	18.2	4
Female	81.8	18
Age (years)		
Range (mean)	25–63 (47.86)	
Population		
Castellon	0	0
Valencia	100	22
Alicante	0	0
Offsprings		
0	27.3	6
1	36.4	8
2	27.3	6
3	9.1	2
Level of studies		
Bachelor’s degree/Licentiate degree	50	11
Master’s degree	9.1	2
Doctoral degree	27.3	6
Others	13.6	3
Subject		
Sciences	45.5	10
Humanities/arts	9.1	2
Others	45.5	10
Professional experience		
Range (mean)	1–34 (18.64)	

AF: Absolute Frequency.

**Table 2 ijerph-20-07087-t002:** Correlations of the AQ-27 variable pre- and post-intervention. Wilcoxon Test.

Variable	Range	Mean	*SD*	Wilcoxon (*p*-Value)
Responsibility pre	1–9	2.75	1.61	0.56
Responsibility post	1–9	2.95	1.13	
Pity pre	1–9	6.09	1.82	0.4
Pity post	1–9	6.20	1.33	
Anger pre	1–9	2.49	1.72	0.001 *
Anger post	1–9	1.47	0.65	
Help pre	1–9	7.19	1.79	0.002 *
Help post	1–9	8.09	1.30	
Danger pre	1–9	3.08	1.40	0.001 *
Danger post	1–9	2.10	1.17	
Fear pre	1–9	2.62	1.66	0.003 *
Fear post	1–9	1.84	1.13	
Segregation pre	1–9	2.10	0.92	0.037 *
Segregation post	1–9	1.56	0.82	
Avoidance pre	1–9	4.00	2.08	0.031 *
Avoidance post	1–9	2.85	1.80	
Coercion pre	1–9	6.62	3.22	0.002 *
Coercion post	1–9	5.26	3.90	

* *p* < 0.05.

**Table 3 ijerph-20-07087-t003:** Correlations between the variables “Age” and “AQ-27 pre-intervention”.

Variable	Range	Mean	*SD*	*r*	*p*-Value
Age	25–63	47.86	10.12	0.28	0.2
Responsibility pre	1–9	2.75	1.61		
Age	25–63	47.86	10.12	−0.163	0.482
Pity pre	1–9	6.1	1.82		
Age	25–63	47.86	10.12	−0.140	0.545
Anger pre	1–9	2.49	1.72		
Age	25–63	47.86	10.12	−0.187	0.417
Danger pre	1–9	3.08	1.40		
Age	25–63	47.86	10.12	−0.484	0.026 *
Fear	1–9	2.62	1.66		
Age	25–63	47.86	10.12	−0.272	0.233
Help	1–9	7.19	1.79		
Age	25–63	47.86	10.12	−0.086	0.709
Coercion	1–9	6.62	3.22		
Age	25–63	47.86	10.12	0.149	0.520
Avoidance	1–9	4.0	2.08		
Age	25–63	47.86	10.12	−0.55	0.812
Segregation	1–9	2.1	0.92		

* *p* < 0.05.

**Table 4 ijerph-20-07087-t004:** Correlations between the variable “Professional experience” and “AQ-27 pre-intervention”.

Variable	Range	Mean	*SD*	*r*	*p*-Value
Experience	1–34	47.86	18.64	0.268	0.240
Responsibility pre	1–9	2.75	1.61		
Experience	1–34	47.86	18.64	−0.201	0.383
Pity pre	1–9	6.1	1.82		
Experience	1–34	47.86	18.64	−0.121	0.601
Anger pre	1–9	2.49	1.72		
Experience	1–34	47.86	18.64	−0.200	0.384
Danger pre	1–9	3.08	1.40		
Experience	1–34	47.86	18.64	−0.460	0.036 *
Fear	1–9	2.62	1.66		
Experience	1–34	47.86	18.64	−0.324	0.153
Help	1–9	7.19	1.79		
Experience	1–34	47.86	18.64	−0.179	0.437
Coercion	1–9	6.62	3.22		
Experience	1–34	47.86	18.64	0.211	0.359
Avoidance	1–9	4.0	2.08		
Experience	1–34	47.86	18.64	−0.001	0.998
Segregation	1–9	2.1	0.92		

* *p* < 0.05.

**Table 5 ijerph-20-07087-t005:** Correlations of “Psychosis test”, “Age”, and “Professional experience”.

Variable	Range	Mean	*r*	*p*-Value
Psychosis test	0–10	4.89	−0.23	0.3
Age	18–75	47.86		
Psychosis test	0–10	4.89	−0.17	0.4
Professional experience	1–60	18.64		

**Table 6 ijerph-20-07087-t006:** Correlations of “AQ-27 pre-intervention” and “Psychosis test”.

Variable	Range	Mean	*r*	*p*-Value
Psychosis test	0–10	4.89	−0.273	0.258
Responsibility pre		2.75		
Psychosis test	0–10	4.89	−0.056	0.819
Pity pre	1–9	6.1		
Psychosis test	0–10	4.89	−0.197	0.420
Anger pre	1–9	2.49		
Psychosis test	0–10	4.89	−0.286	0.235
Danger pre	1–9	3.08		
Psychosis test	0–10	4.89	−0.217	0.371
Fear	1–9	2.62		
Psychosis test	0–10	4.89	0.514	0.024 *
Help	1–9	7.19		
Psychosis test	0–10	4.89	−0.176	0.472
Coercion	1–9	6.62		
Psychosis test	0–10	4.89	−0.295	0.220
Avoidance	1–9	4.0		
Psychosis test	0–10	4.89	−0.049	0.841
Segregation	1–9	2.1		

* *p* < 0.05.

**Table 7 ijerph-20-07087-t007:** Mean scorings of the satisfaction survey.

Contents	Methodology	Material	Evaluation	Overall	Teachers
8.5	9	8.5	8.7	9	9.2

## Data Availability

Data available on request due to privacy and ethical restrictions. Data presented in this study are available upon request from the corresponding author.

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
