# Peer review of "The Promotion of Mental Health and Prevention of First-Episode Psychosis: A Pilot and Feasibility Non-Randomised Clinical Trial"

_ijerph, 2023, doi:10.3390/ijerph20227087_

Round 1

Reviewer 1 Report (Previous Reviewer 1)

Comments and Suggestions for Authors

This is the second time I am evaluating this project, as the authors have made a commendable effort to improve it and deserve a second revision, so I have started by revisiting my initial evaluation and recovering the reasons that led me to initially reject its publication and see if they have been sufficiently improved in this second version.

Unfortunately, I believe that the improvement has not been sufficient, as the main shortcomings are still present.

The main reasons for the initial rejection were: lack of a control group, lack of follow-up, small sample size and insufficient intervention to bring about a sustained change in attitudes over time.

All these shortcomings are still there to the same extent so my opinion does not change and I still think that this manuscript should not be published in a journal of this impact level. However, I still think (now that the improvements have been made, all the more so) that this project can be published in a journal of lower impact and less demanding standards, unless the authors decide to make the improvements that will make up for these shortcomings.

With respect to the rest of the issues I first highlighted, I see that all of them are still present to a lesser extent, but the improvements have not been sufficient.

Author Response

Dear reviewer, thank you very much for your contributions. Undoubtedly, it has made us improve our research, as we have learned from each of the recommendations you have made. Thank you very much for your sincerity, for your time and for the improvements you have proposed. If the article has been improved, it has been thanks to you.

The aspects indicated are the result of the fact that the initial design is a pilot study, with reference to sample size, follow-up, and control group. A pilot study, by definition, is conducted on a small sample to test the feasibility of implementation and to take into account more complex methodological aspects for further research. We have taken note so that, in future research, and in the context of a study with a group of representatives (non-pilot) subjects, it will be carried out in the way you have indicated. In this case, it is a pilot study, and this is the reason these changes have not been made. Nor have we intended to bias or alter the research, as this is how it was conducted from the beginning, and we are faithful to science. Thus, our research is understood from that point of view, as a pilot study. With your contributions, we think that our research is the starting point for another more complex design, which will be conducted with a control group, with a larger sample and with an adequate follow-up. Once again, thank you for all your work, please accept the gratitude we send you from Valencia, Spain. Best regards.

Reviewer 2 Report (Previous Reviewer 3)

Comments and Suggestions for Authors

The authors have followed up the recommendations and suggestions from reviewers. The paper has been substantially improved. I consider that the paper can be published in its current form.

Author Response

Thank you very much for your comments, no doubt with your contributions the article has improved a lot and has given us the opportunity to learn in this world of research. Thank you again. Best regards.

This manuscript is a resubmission of an earlier submission. The following is a list of the peer review reports and author responses from that submission.

Round 1

Reviewer 1 Report

Comments and Suggestions for Authors

This research project has serious gaps and needs too many improvements to be published, so I cannot recommend its publication, but rather that it be substantially improved and submitted to another journal. The main reasons are as follows:

The presentation of the manuscript should be monitored and care should be taken to ensure that the spacing between paragraphs and the formatting, style and font of the text is uniform. The text has minor flaws in this respect, which should be corrected.

The introductory section has an adequate structure, but is rather superficial, lacking the analysis of previous similar studies. This introduction should be expanded with a description at least of the results of 2-5 previous similar studies in order to adequately contextualise the research carried out.

The introduction does not say anything about the role of education and teachers, despite the fact that they are the sample chosen for the study. This is an important gap that needs to be filled.

The manuscript has important methodological gaps: it does not describe the conditions under which the pre- and post-treatment tests are carried out, it does not describe the intervention, it only makes one post-treatment measurement without any follow-up or information on the second measurement, the sample is small, some tests lack reliability and validity and therefore their use in inferential statistical analyses is not recommended, effect sizes are not shown in the analyses, the use of Pearson's correlation is not justified over Spearman's correlation (more appropriate for ordinal variables), etc.

It is very debatable whether attitudes can be changed with a 3-hour programme, but if there is no follow-up, it cannot be established that the results are stable and it cannot be known whether they are generalisable as there is a very specific and small sample.

The intervention should be specified concretely, as its benefits are evaluated, but very little is known about what it consists of.

The discussion compares the results with studies that should have been described in the Introduction.

For all these reasons, I consider that this manuscript is not recommended for publication.

Author Response

Responses to reviewer 1 (in red)

This research project has serious gaps and needs too many improvements to be published, so I cannot recommend its publication, but rather that it be substantially improved and submitted to another journal.

The main reasons are as follows:

The presentation of the manuscript should be monitored and care should be taken to ensure that the spacing between paragraphs and the formatting, style and font of the text is uniform. The text has minor flaws in this respect, which should be corrected.

Formatting is unified throughout the document with Calibri font (body), 11. Centered alignment and 1.0 line spacing.

The introductory section has an adequate structure, but is rather superficial, lacking the analysis of previous similar studies. This introduction should be expanded with a description at least of the results of 2-5 previous similar studies in order to adequately contextualise the research carried out.

Question answered in the May review. The theoretical framework has been updated and completed deepening in studies carried out in the last few years (references from 2020, 2021 and 2022). plans and programmes of health promotion with a 2026 projection have been added. There are few similar articles conducted on high school teachers and evidence for the purpose of inference. This is the reason why it is a pilot study.

 The introduction does not say anything about the role of education and teachers, despite the fact that they are the sample chosen for the study. This is an important gap that needs to be filled.

The relationship between adolescents, teachers and stigma is described in the following sections:

As a matter of fact, there is evidence that states that adolescents with mental disorders have experienced stigmatisation from their peers, families, and teachers, allowing negative emotional reactions, self-discriminatory attitudes, and isolation (Amado-Rodríguez et al., 2022; Chang et al., 2022). This fact is particularly striking since teachers can be a unique figure for providing the necessary support in these cases (Yamaguchi et al., 2020).

Given that psychosis, or First-Episode Psychosis (FEP) if it occurs for the first time, usually debuts from age 15 and its onset is progressive and insidious (Geng et al., 2018; M. Millan et al., 2016; Pardo-de-Santayana et al., 2020), it would be interesting to perform a psychoeducational programme to increase knowledge on mental health and FEP on teachers and counsellors of a secondary school in order to reduce the associated stigma. Some of the ways to carry it out that have been documented in the last few years have been psychoeducation and live interpersonal exchange of people with experiences on mental health disorders (Yin et al., 2020), although similar effects have been found in the latter when people contact virtually (videos, web interactions) (Gronholm et al., 2017; Tergesen et al., 2021).

The manuscript has important methodological gaps: it does not describe the conditions under which the pre- and post-treatment tests are carried out, it does not describe the intervention, it only makes one post-treatment measurement without any follow-up or information on the second measurement, the sample is small, some tests lack reliability and validity and therefore their use in inferential statistical analyses is not recommended, effect sizes are not shown in the analyses, the use of Pearson's correlation is not justified over Spearman's correlation (more appropriate for ordinal variables), etc.

Question answered in the May review. Complementary information related to the intervention was added, since it was a psychoeducational program based on the recommendations of national and international organizations: identification of risk and protective factors for mental health, demystification of information, alarm symptoms, through audiovisual material, games, and a recorded interview with a person with a mental health diagnosis.

Both the program and the questionnaires were carried out at the educational center itself.

The Stigma Attribution Scale (AQ-27) is validated in Spanish with a cronbach's alpha of 0.855 (indicated in the methodology section); the test assessing general knowledge about mental health was carried out by experts in the field using the Delphi technique (also indicated in the article) since there is no validated scale that assesses such knowledge.

The sample is appropriate for a pilot study. And given that it is a non-parametric sample, the Wilcoxon test was chosen and p-values of less than 0.05 and even 0.001 (statistically significant at a 99% CI) were obtained.

Pearson's correlation was used to analyze the relationship between continuous quantitative variables. Since the level of stigma was not categorized by ranges (mild-moderate-high), Spearman's test was not applied.

In relation to the intervention. Question answered in the May review. the intervention has been properly described, with more detailed content. Date of performance of the intervention has been specified, as well as the procedure of randomisation of centres and the election of individuals. The sample and its representation have not been determined, since it is a pilot study.

In relation to the sample. Question answered in the May review. the sample is appropriate for a pilot study.

It is very debatable whether attitudes can be changed with a 3-hour programme, but if there is no follow-up, it cannot be established that the results are stable and it cannot be known whether they are generalisable as there is a very specific and small sample.

Initially the intervention was planned to be carried out in 6 hours but given that the time in contact with the subjects was limited due to their work activity, the time was reduced considering that the objectives of the study were met. Finally, it was evidenced that the reduction of time did not affect the results, and even the stigmatizing attitudes were modified and reduced. As a continuation of this study, we are working on a prospective follow-up of the subjects to analyze whether the levels of stigma change over time.

The intervention should be specified concretely, as its benefits are evaluated, but very little is known about what it consists of.

The intervention consisted of two parts: stigma (definition, causes and consequences; reflection and eradication), and “psychoeducation in psychosis” (risk factors related to mental disorders, conflict resolution, substance abuse prevention, early detection of alarm symptoms, referral to specialised services, treatment, recovery in school).

The methodology implemented, the number of sessions and the time during which it was applied are indicated.

The discussion compares the results with studies that should have been described in the Introduction. For all these reasons, I consider that this manuscript is not recommended for publication

The research is new, so there is little scientific evidence on the subject. In any case, the inference has been made with similar interventions that have been able to provide relevant information in this regard.

Reviewer 2 Report

Comments and Suggestions for Authors

All corrections should be done before start of publication process

There is no highlights----should be

The abstract needs more improvement and should contained backgrounds(including aims) , Methods , Results and conclusion

Add legal instead of legislative

Add psychosis, Mental disorders and health to the keywords

Add diagnostic instead of stigma

Introduction is very long ----why ??? be more summarize

Prevention of pathology -----explain the pathogenesis and which alterations do you mean ????

WHO-recommendations----write the year and issue number

Environments-stigma----clear the relationship between then

Add reference after stigma

Clear the connections between adolescents , mentaldisorders and stigmatization ? How can improve the condition(recommendations)

The most used methodology without references ??/

There is no reference for the statistical analysis

Quasi-test ---add reference

There is no plan for the study area

There is no charts or figures ---why ???

A copy of questionnaire should be enclosed

Discussion is very long ----be more concise

There is no data availability

There is no author contributions

There is no recommendation

write as Table(1):------------ and Fig.(1):-----------etc

Some cited references need to be more updated

Some journal names were written abbreviated , while others were not ---why ??? same style should be ---APPLY FOR ALL

Ref(4)---delete (internet)

Ref(5)---rewrite again

Ref(10)----delete vol.

Ref(18/45/49/51----of a missing data either issue, volume, number or pages ---Apply for all

Ref(48)---write as 56(2):243-255,2019----instead of others

Comments on the Quality of English Language

Good 

Author Response

Responses to reviewer 1 (in red)

The quality and the scientific soundness is some extend okay 9 authors /one paper---!!!???

The number of authors has been reduced to three, in view of the recommendations.

Serial numbers per each pages ---makes the peer review process very difficult

All corrections should be done before start of publication process There is no highlights----should be

The abstract needs more improvement and should contained backgrounds(including aims) , Methods , Results and conclusion

The summary has been structured by sections. The objectives have been added.

Add legal instead of legislative

The recommendation to change "legal" to "legislative" has been followed.

Add psychosis, Mental disorders and health to the keywords Add diagnostic instead of stigma

The recommended keywords are added.

Introduction is very long ----why ??? be more summarize

The introduction consists of 680 words with the objective of understanding the characteristics of adolescence, the onset of mental health disorders, the current national and international plans, and the need for practical programs for teachers.

Prevention of pathology -----explain the pathogenesis and which alterations do you mean ????

Information related to the clinical presentation of a first psychotic episode has been introduced in paragraph 6.

WHO-recommendations----write the year and issue number

WHO recommendations are referenced in the text and in the bibliography. They refer to the Mental health action plan 2013-2020.

Environments-stigma----clear the relationship between then

In fact, there is evidence that states that adolescents with mental disorders have experienced stigmatisation from their peers, families, and teachers, allowing negative emotional reactions, self-discriminatory attitudes, and isolation (Amado-Rodríguez et al., 2022; Chang et al., 2022). This fact is particularly striking since teachers can be a unique figure for providing the necessary support in these cases (Yamaguchi et al., 2020).

At the same time, and despite the multiple instructions of carrying out specific programmes to increase knowledge on mental health and reduce stigma associated to the health, social and educational systems, there is scarce scientific evidence on the latter, since the majority of studies made have focused in students of different categories or in healthcare professionals (Beaulieu et al., 2017; Fernandes et al., 2022; Ma & Hsieh, 2020; Sheehan & Corrigan, 2020; Sickel et al., 2019; Walsh & Foster, 2021). Moreover, the few programmes performed on teaching staff have dealt almost exclusively with disorders like anxiety and depression (Holder et al., 2019; Jorm et al., 2010), leaving aside important disorders like psychosis, due to its high rates of Disability-Adjusted Life Years (DALY) (Ministry of Health, 2022; Tejado et al., 2022).

Add reference after stigma

The following references are added

Clear the connections between adolescents , mentaldisorders and stigmatization ? How can improve the condition(recommendations)

The following information is added:

Implications for the profession:  There are adolescents who have suffered stigma from their teachers, and consequently have minimized the symptoms and have not asked for help. For this reason, we have implemented a nursing intervention, based on education and promotion of mental health with the aim of expanding knowledge and reducing stigma. In fact, this intervention, which we have carried out on high school teachers, has managed to reduce most stigmatizing attitudes measured on the stigma attribution scale.

The most used methodology without references ??

In the section explaining the intervention, the methodology used is specified (psychoeducation, “health Education and Promotion framework, audiovisual materials, games, websites and a recorded interview with a person that suffers from schizoaffective disorder were used, since they are resources that have shown to be effective to reduce stigma (Tippin & Maranzan, 2019; Williams et al., 2021); in addition, it seems like the latter is the one that has made it possible to obtain better results (Corrigan et al., 2012; Fong & Mak, 2022).” (Amado-Rodríguez et al., 2022; Generalitat Valenciana, 2016, 2019; Ministry of Health, 2022)

There is no reference for the statistical análisis

SPSS version 22 was used for statistical analysis.

Quasi-test ---add reference

The test was conducted by experts associated with a functional unit of first psychotic episodes through the Delphi technique with the aim of creating a series of questions that would objectify the general mental health knowledge of the subjects.

There is no charts or figures ---why ???

Tables have been included to make the information clearer and more visual, but they could be reduced if recommended by the reviewers and the editor.

A copy of questionnaire should be enclosed

The questionnaire is attached at the request of the reviewers.

There is no data availability

The data are kept in the custody of the health department where the study was conducted and are legally available to other researchers.

 There is no author contributions

Se añaden las contribuciones de los autores en el “Title page”

  • Lucía SANTONJA-AYUSO: santonja@gmail.com

Orcid: 0000-0002-8269-5787

Mental Health Nurse. Department of Nursing, Faculty of Health Sciences, Universitat Jaume I, Avda. Sos I Baynat s/n, 12071 Castellón, Spain.

She has participated in the Conceptualization of the study. Moreover she has conducted the literature review, designing the search strategy, analysing data and writing the paper.

  • José Vicente CARMONA SIMARRO: carmona@universidadeuropea.es Corresponding author

Orcid: 0000-0003-4550-0685

PhD. Department of Nursing, European University of Valencia. Spain. c/Paseo Alameda, 7, 46010 Valencia, Spain.

He has participated in the research process, has supervised the study, and review the manuscript drafts.

  • Laura ANDREU-PEJÓ: pejo@uji.es

+34627632871

Orcid: 0000-0002-2944-9878

PhD. Faculty of Health Sciences, Universitat Jaume I, Avda. Sos I Baynat s/n, 12071 Castellón, Spain.

She has participated in the Conceptualization of the study; the methodology designed and supervised the whole process.

write as Table(1):------------ and Fig.(1):-----------etc

Reviewers' recommendations are updated

Some cited references need to be more updated

The bibliographic references have been updated and most of them are from documents of the last five years (maximum scientific evidence to make the inference in the discussion). Occasionally, and with the aim of making the introduction, especially the contextual part, articles of more than 5 years of interest for the research have been incorporated.

 Some journal names were written abbreviated , while others were not -- -why ??? same style should be ---APPLY FOR ALL

Bibliographic references have been made according to the recommendations of the reviewers, to unify criteria.

 Ref(4)---delete (internet): sigue en internet; se añade URL: https://irep.ntu.ac.uk/id/eprint/46099/1/1536568_Griffiths.pdf

Ref(5)---rewrite again: escrito según las recomendaciones y se añade URL: https://psycnet.apa.org/record/2018-35597-000

Ref(10)----delete vol. escrito según las recomendaciones. Se añade URL: https://psycnet.apa.org/record/2021-11456-001

Ref(18/45/49/51----of a missing data either issue, volume, number or pages ---Apply for all

Ref(48)---write as 56(2):243-255,2019----instead of others

Reviewer 3 Report

Comments and Suggestions for Authors

This is a very interesting study investigating the effects of a specific intervention for the promotion of mental health and prevention of a first episode of psychosis. The paper is well-written and of interest for the journal; however, several changes are recommended before considering it for publication.

ABSTRACT

1- It is not necessary to extend the explanation about the psychiatric care in Spain. The first part of the abstract should be focus on explaining what are specific interventions for the promotion and prevention.

2- Please, define what AQ-27 is, before considering the abbreviation.

3- The main aims of the paper should be described in the abstract section before the description of the study design (Non-randomized clinical trial).

INTRODUCTION

1- I recommend to add some more references about promotion and prevention of mental health in people potentially suffering for first episode of psychoses. 

METHODS

1- I recommend to rename "Tools" as "Assessment instruments" or "psychometric instruments". 

RESULTS

1- How many men and women were included? Please, describe the sample size of both subgroups (18% men, 82% women), at the beginning of the description of the patients.

2-"Masculine" and "femenine" should be changed to "Male" and "female" or men and women.

DISCUSSION

1-Please, avoid the term "coinciding" and apply "in line with", "in agreement with", etc.

2- In the conclusion section, the authors should avoid to summarize the results.

Author Response

Responses to reviewer 1 (in red)

 ABSTRACT 1- It is not necessary to extend the explanation about the psychiatric care in Spain. The first part of the abstract should be focus on explaining what are specific interventions for the promotion and prevention

The abstract has been modified and updated according to the recommendations of reviewers 2 and 3.

2- Please, define what AQ-27 is, before considering the abbreviation.

AQ-27 is defined before the abbreviation according to the reviewer's recommendation.

3- The main aims of the paper should be described in the abstract section before the description of the study design (Non-randomized clinical trial).

The main objectives of the study are described in the abstract, followed by a description of the study design.

INTRODUCTION 1- I recommend to add some more references about promotion and prevention of mental health in people potentially suffering for first episode of psychoses.

Reference has been made to national and international reference bodies in the field of mental health, which recommend the implementation of anti-stigma programs in the classroom.

METHODS 1- I recommend to rename "Tools" as "Assessment instruments" or "psychometric instruments".

The section has been renamed following the recommendations of Reviewer 3.

RESULTS

1- How many men and women were included? Please, describe the sample size of both subgroups (18% men, 82% women), at the beginning of the description of the patients.

The description of the sample can be found in the first section of the results, together with Table (1) providing both the percentages and the absolute frequencies of the men and women in the study.

2-"Masculine" and "femenine" should be changed to "Male" and "female" or men and women

DISCUSSION

1-Please, avoid the term "coinciding" and apply "in line with", "in agreement with", etc

Se ha cambiado conforme a la recomendación.

2- In the conclusion section, the authors should avoid to summarize the results

The conclusions were based on the general and specific objectives of the study.